



# The Lehtinen-Pirjola Method Modified for Efficient Modelling of Geomagnetically Induced Currents in Multiple Voltage Levels of a Power Network

Risto J. Pirjola[1], David H. Boteler[1], Loughlin Tuck[1,2], Santi Marsal[3]

[1] Geomagnetic Laboratory, Natural Resources Canada, Ottawa, Ontario, Canada
[2] Defence Research and Development Canada, Dartmouth, Nova Scotia, Canada
[3] Observatori de l'Ebre (OE), Univ. Ramon Llull - CSIC. Horta Alta, 38. 43520 Roquetes (Spain)

*Correspondence email for proofs: david.boteler@canada.ca*

**Abstract.** The need for accurate assessment of the geomagnetic hazard to power systems is driving a requirement to model geomagnetically induced currents (GIC) in multiple voltage levels of a power network. The Lehtinen-Pirjola method for modelling GIC is widely used but was developed when the main aim was to model GIC in only the highest voltage level of a power network. Here we present a modification to the Lehtinen-Pirjola (LP) method designed to provide an efficient method for modelling GIC in multiple voltage levels. The LP method calculates the GIC flow to ground from each node. However, with a network involving multiple voltage levels many of the nodes are ungrounded, i.e. have infinite resistance to ground which is numerically inconvenient. The new modified Lehtinen-Pirjola (LPm) method replaces the earthing impedance matrix [Ze] with the corresponding earthing admittance matrix [Ye] in which the ungrounded nodes have zero admittance to ground. This is combined with the network admittance matrix [Yn] to give a combined matrix ([Yn]+[Ye]), which is a sparse symmetric positive definite matrix allowing efficient techniques, such as Cholesky decomposition, to be used to provide the nodal voltages. The nodal voltages are then used to calculate the GIC in the transformer windings and the transmission lines of the power network. The LPm method with Cholesky decomposition also provides an efficient method for calculating GIC at multiple time steps. Finally, the paper shows how software for the LP method can be easily converted to the LPm method and provides examples of calculations using the LPm method.

## 1 Introduction

Geomagnetic disturbances produce geoelectric fields that drive geomagnetically induced currents (GIC) in power networks. These GIC flow along transmission lines and through transformer windings where they can cause half-cycle saturation leading to harmonic generation, increased consumption of reactive power and transformer heating. These, in turn, can cause misoperation of protective relays and voltage sag and, in extreme cases, damage to transformers and system collapse (Kappenman and Albertson, 1990; Bolduc, 2002; Molinski, 2002; Kappenman, 2007; Guillon et al., 2016). A key requirement





for understanding the impact of geomagnetic disturbances on power networks is the ability to model the GIC produced in a network by specified geoelectric fields. In 1985, Lehtinen and Pirjola published a landmark paper that provides the first description of a stand-alone method for modelling GIC. The Lehtinen-Pirjola (1985) method (hereafter referred to as the 'LP method') has been widely used in the geophysics community and provided the basis for GIC studies in many countries (e.g.

Pirjola and Lehtinen, 1985; Mäkinen, 1993; Mäkinen et al., 1993; Thomson et al., 2005; Wik et al., 2008; Viljanen et al., 2012; Torta et al., 2014; 2017; Divett et al., 2018).

The LP method was designed at a time when mostly only the highest voltage levels of a power network were considered in GIC calculations. This was because the transmission lines at the lower voltage levels have higher resistance so will experience smaller GIC values. However, in a desire to provide more comprehensive modelling of GIC in a power network, many modern

studies are now looking to model GIC in multiple voltage levels in a power network. The LP method has been effectively used for such studies (e.g., Mäkinen, 1993; Mäkinen et al., 1993; Viljanen et al., 2012; Divett et al., 2018); however, using the LP method for multiple voltage levels involves many ungrounded nodes, thus having infinite resistance to ground, which is numerically inconvenient. Also, the main focus of the LP method was the GIC flow to ground through the transformer primary windings, which was the desired output when modelling a single voltage level of a power network. However, models for

multiple voltage levels require calculation of the nodal voltages which are then used to calculate the GIC in the transformer windings (Boteler and Pirjola, 2014).

In this paper we show how the LP method can easily be modified to efficiently model GIC in multiple voltage levels of a power network by converting the LP method to calculate the nodal voltages directly. First we summarise the steps in the LP method and then show how these can be modified to give the modified Lehtinen-Pirjola method (hereafter referred to as the

'LPm method'). We also show that the LPm method involves inversion of a matrix that is symmetric positive definite allowing the use of efficient methods including sparse matrix techniques. Then we show how software for GIC calculations using the LP method can easily be converted to the LPm method and provide example calculations for the benchmark model introduced by Horton et al. (2012), including tables of values at intermediate steps, to help people transitioning their modelling from the LP method to the LPm method.

## 55  2 Lehtinen-Pirjola Method

The GIC modelling method derived by Lehtinen and Pirjola (1985), the 'LP' method, is derived by starting with Kirchhoff's current law that the net current flowing into a node, $k$, on branches from other nodes, $n$, is equal to the current flowing to ground from node $k$ (LP equation 8):

$$i_k = \sum_{n=1}^{N} i_{nk} = -\sum_{n=1}^{N} i_{kn}$$


(1)



and relate the current in a branch $i_{kn}$ to the driving electromotive force (emf) $e_{kn}$ (if there is one), the voltage difference between the nodes at the ends of the line $v_k$ and $v_n$, and the admittance of the branch (LP equation 7) $y_{kn}$:


$$i_{kn} = y_{kn}[e_{kn} + (v_k - v_n)] \tag{2}$$

Substituting (2) into (1) gives (LP equation 9):

$$i_k = -\sum_{n=1}^{N} y_{kn}[e_{kn} + (v_k - v_n)] \tag{3}$$

Note that, when considering multiple voltage levels, branches in the network consist of not just transmission lines but also transformer windings. The transmission lines experience the driving emf produced by the magnetic field variations, whereas the transformer windings do not.

The driving emf in each transmission line is represented by an equivalent current source

$$j_{kn} = e_{kn} y_{kn} \tag{4}$$


The equivalent current sources are then summed to give the current source directed into each node (LP equation 13).

$$J_k^e = -\sum_{\substack{n=1 \\ n \neq k}}^{N} j_{kn} . \tag{5}$$

Making this substitution in (3) gives


$$i_k = J_k^e - \sum_{\substack{n=1 \\ n \neq k}}^{N} (v_k - v_n) y_{kn} , \tag{6}$$

Thus

$$i_k = J_k^e - v_k \sum_{\substack{n=1 \\ n \neq k}}^{N} y_{kn} + \sum_{\substack{n=1 \\ n \neq k}}^{N} v_n y_{kn} , \tag{7}$$

The first summation represents the dependence of current $i_k$ on voltage $v_k$ so gives diagonal elements of a network admittance 85 matrix

$$Y_{kk}^n = \sum_{\substack{n=1 \\ n \neq k}}^{N} y_{nk} , \tag{8}$$





The second summation represents the dependence of current $i_k$ on all the other nodal voltages $v_n$ so gives the off-diagonal elements of the (symmetric) network admittance matrix

$$Y^n_{kn} = -y_{kn} \qquad n \neq k \tag{9}$$

Combining the above equations gives (LP equation 11):

$$i_k = J^e_k - \sum_{n=1}^{N} v_n Y^n_{kn} \tag{10}$$

This can be written in matrix form

$$\left[ I^e \right] = \left[ J^e \right] - \left[ Y^n \right]\left[ V^n \right] \tag{11}$$

Where the elements of column matrix $[I^e]$ are the currents $i_n$. and the elements of column matrix $[V^n]$ are the voltages $v_n$.

LP make the substitution

$$\left[ V^n \right] = \left[ Z^e \right]\left[ I^e \right] \tag{12}$$

where $\left[ Z^e \right]$ is the earthing impedance matrix. Thus

$$v_k = \sum_{n=1}^{N} Z^e_{kn} i_n \tag{13}$$

Substituting (12) into (11) gives a matrix equation involving only the node to ground currents $I^e$ as the unknowns

$$\left[ I^e \right] = \left[ J^e \right] - \left[ Y^n \right]\left[ Z^e \right]\left[ I^e \right] \tag{14}$$

Gathering terms in $\left[ I^e \right]$ gives

$$\left( [I] + \left[ Y^n \right]\left[ Z^e \right] \right)\left[ I^e \right] = \left[ J^e \right] \tag{15}$$

where $[I]$ is the unit matrix with size equal to the number of nodes in the model network. Equation (15) can be solved by matrix inversion to give the currents flowing to ground (LP equation 12)

$$\left[ I^e \right] = \left( [I] + \left[ Y^n \right]\left[ Z^e \right] \right)^{-1} \left[ J^e \right] \tag{16}$$



The values of $[I^e]$ were the desired output when modelling a single voltage level of a power network. However, if there was more than one transformer at a substation (as usually occurs), it was necessary to split the current in proportion to the admittances of the transformer windings to determine the fraction of the current that flowed in each transformer winding.

Now, when modelling the GIC in multiple voltage levels of a power network, many of the nodes are ungrounded. However, the LP method needs to specify an earthing impedance for each node. This is done by adding 'virtual' connections to ground from each ungrounded node (Mäkinen, 1993; Pirjola, 2005). These virtual earthing connections have infinite resistance, but this cannot be represented in the earthing impedance matrix $\left[ Z^e \right]$, so a high value is used instead. The LP method then gives the current flow to ground from each node, including small current values through the virtual earthing connections. It is necessary to use the $\left[ I^e \right]$ values and the earthing impedance $[Z^e]$ in equation (12) to calculate the nodal voltages $[V^n]$. The nodal voltages are then used to calculate the GIC flow in the branches using equation (2). This is the equation to use for the GIC in the transmission lines. For branches of the network that are transformer windings, there is no driving emf so equation (2) reduces:

$$i_{kn} = y_{kn}(v_k - v_n) \tag{17}$$

to give the GIC flow in the transformer windings.

### 3 Lehtinen-Pirjola Modified Method

When modelling GIC in multiple voltage levels of a power network, it is necessary to calculate the nodal voltages before calculating the GIC in the transmission lines and transformer windings. In the Lehtinen-Pirjola modified (LPm) method the matrix equations are modified to provide a solution in terms of the nodal voltages. This also has the advantage that there is no need to add virtual earthing connections to ground from the ungrounded nodes.

To convert the currents flowing to ground $[I^e]$ provided by the LP method to nodal voltages $[V^n]$ start with equation (15) and make the substitution from equation (12) (Pirjola, 2007)

$$\left[ I^e \right] = \left[ Z^e \right]^{-1} \left[ V^n \right] \tag{18}$$

which gives

$$\left[ J^e \right] = \left( \left[ Z^e \right]^{-1} + \left[ Y^n \right] \right) \left[ V^n \right] \tag{19}$$




The LP method allows for the $[Z^e]$ matrix to have off-diagonal elements representing the voltage produced at node $i$ by currents flowing to ground from other nodes (Pirjola, 2008). However, if the circuit is set up with a node at the neutral point of each substation, this does not happen (see Boteler and Pirjola, 2014). In this case, $[Z^e]$ becomes diagonal with elements equal to the earthing resistances $r_i$ of the nodes and the inverse of $[Z^e]$ is simply the earthing admittance matrix $[Y^e]$ given by


$$Y_{ii}^e = y_i = 1/r_i$$

(20)

$$Y_{ij}^e = 0 \qquad j \neq i$$

Then equation (19) can be rewritten as:

$$\left[ J^e \right] = \left( \left[ Y^e \right] + \left[ Y^n \right] \right) \left[ V^n \right]$$

(21)


The voltages of the nodes are then found by taking the inverse of the sum of the admittance matrices and multiplying by the nodal current sources

$$[V^n] = ([Y^e] + [Y^n])^{-1}[J^e]$$

(22)


These node voltages can then be substituted into (2) and (17) to give the GIC in the transmission lines and the transformer windings. The GIC flow to ground is simply given from Ohm's law using the neutral point node voltage and the admittance to ground (equation (18) with $[Z^e]^{-1} = [Y^e]$).

The LPm method involves inversion of a matrix ($[Y^e] + [Y^n]$) which is symmetric (i.e., Hermitian as the elements are real) and positive definite and can thus be solved using a particularly efficient case of lower-upper (LU) decomposition, the Cholesky decomposition (Press et al., 2007). Note that most of the nodes within the network are unconnected, meaning that $[Y^n]$ has many zeros. This is also the case with $[Y^e]$, so the Cholesky decomposition enables the use of sparse matrix methods (Stott and Alsaç, 1987; Press et al, 2007), thus providing an efficient way to model GIC in multiple voltage levels of a power network.

**4 Calculation of GIC Time Series**

   GIC modelling is now being used, not just to assess the GIC for specified electric field values, but also to determine the variation of GIC throughout a geomagnetic disturbance. If the network configuration does not change during that time (not always the case), then the matrix inversion does not need to be recalculated at every time step.

   If the electric field is assumed to be uniform across the network then linear superposition can be used to calculate the GIC. (A

uniform electric field would be produced, e.g., if calculations are made using data from a single magnetic observatory and a





one-dimensional (1-D) earth conductivity model.) The GIC modelling can be made for two cases: i) a northward electric field of 1 V/km, and ii) an eastward electric field of 1 V/km. For each location $k$ in the network, this modelling gives reference GIC values, $\alpha_k$ and $\beta_k$ for the northward and eastward electric fields that can be scaled by the actual electric field values at each time step and then combined to give the time series of GIC values at that location.


$$i_k(t) = \alpha_k E_N(t) + \beta_k E_E(t) \tag{23}$$

This concept can be extended for using two magnetic observatories (Boteler et al., 2014), but this still requires use of a single 1-D earth conductivity model for the whole network.

In practice there is considerable variability in the earth conductivity structure across a power network. There are many modelling techniques for calculating the electric fields in such cases, ranging from use of multiple 1-D earth models (Marti et

al., 2014) to use of magnetotelluric transfer functions and 3-D earth conductivity models (Weigel, 2017). In these cases, the electric fields across the network can change from place to place and from one time step to the next. This will result in a different set of nodal current sources $[J^e]$ for each time step. However, for much of the time the network configuration may be unchanged, thus once the inverted matrix $([Y^n]+[Y^e])^{-1}$ has been calculated it does not need to be recalculated at each time step and can be used with the nodal current source $[J^e]$ for each timestep to calculate the nodal voltages $[V^n]$ and hence the time

series of GIC values.

However, for GIC calculations using the LPm method even more efficient time series calculations are possible. The solution of a matrix equation, such as equation (21) can be accomplished using LU decomposition, as explained in Press et al. (2007). This involves writing the matrix $([Y^n]+[Y^e])$ as a product of two matrices:

$$[L].[U] = ([Y^n] + [Y^e]) \tag{24}$$

Where $[L]$ is a lower triangular matrix and $[U]$ is an upper triangular matrix.

For a positive-definite symmetric matrix, as is obtained with the LPm method, the $[L]$ matrix can be chosen such that the $[U]$ matrix is the transpose of $[L]$. In this case we can write (24) using the Cholesky decomposition

$$[L].[L^T] = ([Y^n] + [Y^e]) \tag{25}$$

This Cholesky decomposition to solve the linear set is


$$([Y^n] + [Y^e]).[V^n] = ([L].[L^T]).[V^n] = [L].([L^T].[V^n]) = [J^e] \tag{26}$$

By first solving for the vector $P$ such that

$$[L].[P] = [J^e] \tag{27}$$

and then solving

$$[L^T].[V^n] = [P] \tag{28}$$

The advantage is that the solution of a triangular set of equations is quite trivial, as equation (27) can be solved by forward substitution and equation (28) can be solved by back substitution. Also, once the Cholesky decomposition has been done, it is possible to solve with as many right-hand sides as required, one at a time. Thus the LPm method provides a much faster and versatile way of calculating a time series of GIC values.





### 5 Comparison between the LP and LPm Methods

To illustrate the differences between the LP method and the LPm method consider the circuit for a substation with a two-winding transformer and three autotransformers as shown in Figure 1. The LP method requires the addition of virtual connections to ground from nodes 1 and 2, as explained above. However, in the LPm method the connection to ground is expressed as an admittance value. For the ungrounded nodes the admittance to ground is zero, which can easily be included in the earthing admittance matrix without having to add virtual connections to the circuit.

The steps involved in calculating GIC in multiple voltage levels of a power network using the LP method and the LPm method are summarised in Figure 2. In the LPm method, because it involves only admittances and calculates the nodal voltages directly there is no need to add virtual connections to ungrounded nodes and then there is no need to convert the currents through the virtual connections to nodal voltages.

### 6 Example Calculation using the LPm method

To illustrate the use of the LPm technique we present the calculation of GIC in the benchmark model of Horton et al. (2012) shown in Figure 3. The following tables will also provide values for testing when converting software from the LP method to the LPm method.

To construct the network admittance matrix [Yn] and the earthing admittance matrix [Ye] it is first necessary to assign node numbers to the buses and neutral points at each substation. The modelling does not depend on any particular choice of node numbers. Here we use the node number assignment shown in Table 1. Note that some buses are not included in the model for calculating the GIC because they are connected to transformer windings in a delta configuration so there is no path for the GIC to flow. Also, there is no neutral point node at substation 7 because there are no transformers at this site as it is just a switching station connecting transmission lines.

The network admittance matrix [Yn] is constructed using the transmission line information and transformer information presented in Tables II and III of Horton et al. (2012). In the network admittance matrix [Yn] the values are given by equations 8 (for the diagonal elements) and 9 (for the off-diagonal elements). Note that the values are presented for a single phase of the power network and it is assumed that the other two phases are identical. The network admittance matrix values for the benchmark model are given in Table 2.

The earthing admittance matrix $[Y^e]$ is constructed using the substation grounding resistance, $r_g$, presented in Table I of Horton et al. (2012). Note that the GIC from all three phases flow through the substation grounding resistance so the voltage drop here is three times that produced by a current from a single phase. To account for this in a single-phase model the earthing admittance is given by

$$y^e = \frac{1}{3r_g} \tag{29}$$





In the earthing admittance matrix, most nodes are not connected to ground, so their earthing admittance values are zero, and there are only non-zero admittance values for the six neutral point nodes. Earthing admittance matrix values for the benchmark model are shown in Table 3.

Note that the general theory is expressed in terms of impedance and admittance which can have reactive components, but, in

practice, at the frequencies applicable to GIC the reactive components are negligible, and the network characteristics can be described as purely resistive or conductive.

The resulting inversion of the matrix gives $([Y^n] + [Y^e])^{-1}$ shown in Table 4.

Horton et al. (2012) consider two cases: a northward electric field of 1 V/km and an eastward electric field of 1 V/km. They give values for the input induced emf in each line and show the calculated output values both in terms of the nodal (bus)

voltages and the GIC (A/phase) in the transmission lines and transformer windings.

The voltage source in each transmission line and the equivalent current source, calculated from equation (4), are shown in Table 5. These current sources are then summed (equation 5) to give the nodal current sources $[J^e]$ shown in Table 6.

The nodal current sources are then combined with the inverted matrix (Equation 22) to give the nodal voltages shown in Table 7. For nodes 1-11 these give the bus voltages shown in Table V of Horton et al. (2012). For nodes 12-18, combining the nodal

voltage and the substation grounding resistance gives the GIC flow to ground for each substation shown in Table VII of Horton et al., (2012).

The nodal voltages substituted into equations (2) and (17) give the GIC values for the transmission lines and transformer windings shown in Tables VI and VIII of Horton et al. (2012).

## 6 Discussion

The above example calculation shows that the LPm method provides GIC values that exactly match those for the benchmark model provided by Horton et al. (2012). The values in Tables 2 to 7 can also be used to check intermediate steps in the software used for the LPm calculations.

Any software developed to model GIC should be able to exactly match the values provided by the Horton et al. (2012) paper.

The results presented in that paper are not an average of modelling results nor an approximation to the correct values, but are the identical values obtained using four different software implementations. However, initial calculations involving the four different software implementations provided similar but slightly different results. Further investigation showed that the origin of the differences was in the way that distances between substations were being calculated in the different implementations. Some versions used formulas based on a spherical earth and some used formulas taking account of its non-spherical shape. It

was then decided to standardise on substation latitudes and longitudes based on the WGS84 ellipsoid model of the Earth which is used by the global navigation satellite system (GNSS) for geolocation. After this, all the calculations gave exactly the same results. To get the source voltage values presented in Table 5 (and hence match the GIC results for the benchmark model) thus requires using the formulas presented in the Appendix of Horton et al. (2012) for calculating distances between substations.





Many people have used the LP method for calculating GIC in the highest voltage level of their power networks. With the
increasing requirement to calculate GIC in multiple voltage levels of a power network it is hoped that the new LPm method
described above will provide an easy way for converting existing LP software. The conversion is a simple process. Just replace
the earthing impedance matrix $[Z^e]$ with the corresponding earthing admittance matrix $[Y^e]$, form the new matrix $([Y^e] + [Y^n])$
and do the matrix inversion. This directly gives the nodal voltages which are required to calculate the GIC in the transmission
lines and transformer windings. There is no need for any "virtual" nodes or connections. Also, it is more efficient as $([Y^e] +$
$[Y^n])$ is symmetric positive definite, so can be solved using Cholesky decomposition, which is a special case of lower-upper
(LU) decomposition (Press et al., 2007) with $U = L^T$ (i.e., the upper triangular factor is the transpose of the lower triangular
factor).

Power networks, on average, have three transmission lines and one or two transformer windings connected to a bus, so a typical
row in the admittance matrix has only five or six non-zero elements, independent of the overall network size. Thus for  larger
networks, where node numbers can be in the thousands, the admittance matrix will have over 99 % of its values equal to zero.
Cholesky factorization takes advantage of this fact by making use of sparse matrix methods (Stott and Alsaç, 1987; Press et
al, 2007), thus additionally reducing memory usage and computation time. To examine how this affects the GIC modelling ,
we performed calculations for two networks using both the LP and the LPm methods. The networks modelled were 1) the
benchmark network of Horton et al. (2012), which has 18 nodes, and 2) the nation-wide Spanish Power Grid operated by Red
Eléctrica de España (REE), which has 1388 nodes. GIC in the 400 kV part of the REE system was considered by Torta et al.
(2014); for this study we include both the 400 and 220 kV levels of the REE network (see Torta et al., 2021 for reference).
Tests we did showed that the LP and LPm methods both produce matrices that are sparse, so there is potential for sparse matrix
techniques to be applicable. Table 8 shows the calculation times and memory usage for GIC calculations using the LP and
LPm methods. These show that memory usage was drastically reduced when using sparse matrix techniques, with the reduction
being more significant with the larger REE network. The time for the matrix inversion is significantly affected, as expected,
by the size of the matrix involved. For the Horton network (18x18 matrix) the change to sparse techniques actually increased
the inversion time. However, the sparse techniques applied to the REE network (1388x1388 matrix) produced an
approximately order of magnitude reduction in inversion time. This reduction in inversion time was greatest for the LPm
method which was nearly an order of magnitude faster than the LP method.
The column 'Inversion time' in Table 8 reflects the time required to compute $[V^n]$ from equations (25), (27) and (28), thus
including the Cholesky decomposition and the forward and back substitutions in the LPm method; note, in consequence, that
it is not strictly an inversion, though we will refer to it as such. Also note that, when referred to LP, this column reflects the
time to compute $[I^e]$ including the decomposition of $M = [1] + [Y^n][Z^e]$. However, M is not even symmetric in the LP
method, so the decomposition of M is indeed an upper-lower (UL) factorization. The difference in speed of the inversion
between the LP and LPm methods is that LPm involves inversion of a symmetric positive-definite matrix which allows the
use of the technique of Cholesky decomposition, that significantly reduces the time of the inversion process (note that LU
requires the determination of more unknowns). The parameters of the calculations presented in Table 8 are obtained from GIC



modelling using programs in Matlab. Specific inversion times and memory usage will vary with the programming language used, but it is expected that the general results presented here will apply regardless of the programming language used.


## 7 Conclusions

We have presented a new version of the LP method, modified for efficient modelling of GIC in multiple voltage levels of a power system. In the LPm method the earthing impedance matrix, $[Z^e]$, is replaced with an earthing admittance matrix, $[Y^e]$ that is added to the LP network admittance matrix $[Y^n]$ to give a new combined matrix $([Y^n] + [Y^e])$ to be inverted.

Multiplication of the inverted matrix $([Y^n] + [Y^e])^{-1}$ (or equivalently recycling its Cholesky decomposition) by the nodal current sources $[J^e]$ provides a direct calculation of the nodal voltages $[V^n]$. These nodal voltages are then used to calculate the GIC in transmission lines and transformer windings.

Guidance is provided for converting software from the LP method to the LPm method and an example calculation using the benchmark model of Horton et al. (2012) is presented to provide a set of values for testing GIC calculation software.

Calculations of GIC using the LPm method involve a matrix that is symmetric positive definite. This enables a solution to be obtained by Cholesky decomposition, (a specific case of LU decomposition) which is numerically more accurate than computing the matrix inversion itself. The factorization of Cholesky decomposition can always be implemented using sparse matrix techniques, speeding up the calculations for large networks.

Thus the LPm method provides an efficient method for calculating GIC in multiple voltage levels in a power network that
provides a valuable tool for assessing the geomagnetic hazard to power systems.

**Author contribution**:

RP and DB planned the work and developed the theory; LT developed the modelling code and generated the example results; SM performed the tests of matrix inversion; RP and DB wrote the manuscript draft; LT and AM reviewed and edited the
manuscript.

**Competing interests**

"The authors declare that they have no conflict of interest."

**Acknowledgments**

This work was performed as part of the Public Safety Geoscience program and Canadian Hazards Information Service of Natural Resources Canada. SM is grateful to the projects CGL2017-82169-C2-1-R "Holistic characterization of GIC in the Iberian Peninsula: from the analysis of magnetospheric and ionospheric currents to the influence of the lithosphere (IBERGIC)" and PID2020-113135RB-C32 "Vulnerability of the Spanish power transmission grid to the hazard posed by GIC
and measurements for validating the models and forecasts", both funded by FEDER/Ministerio de Ciencia, Innovación y Universidades – Agencia Estatal de Investigación. SM would like to thank Red Eléctrica de España (REE) for supporting this



study. The research that led to these results was in part carried out using funds from "la Caixa" Foundation.   Natural Resources

Canada contribution number 20210276.

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

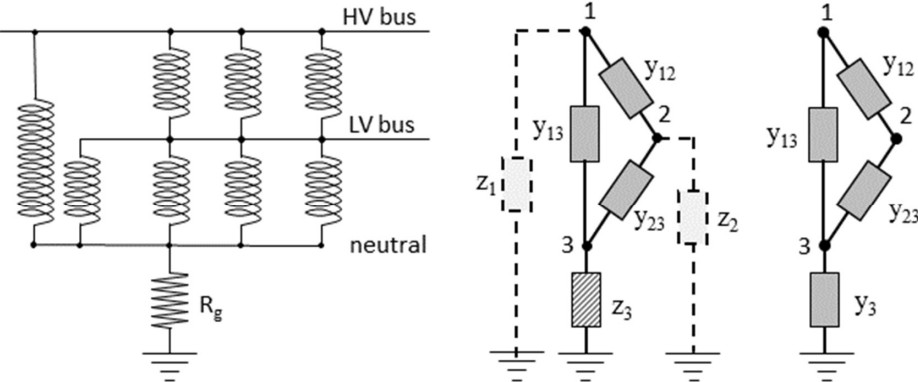

Figure 1: Substation with a two-winding transformer and three autotransformers

and the equivalent circuits for the LP and LPm methods.







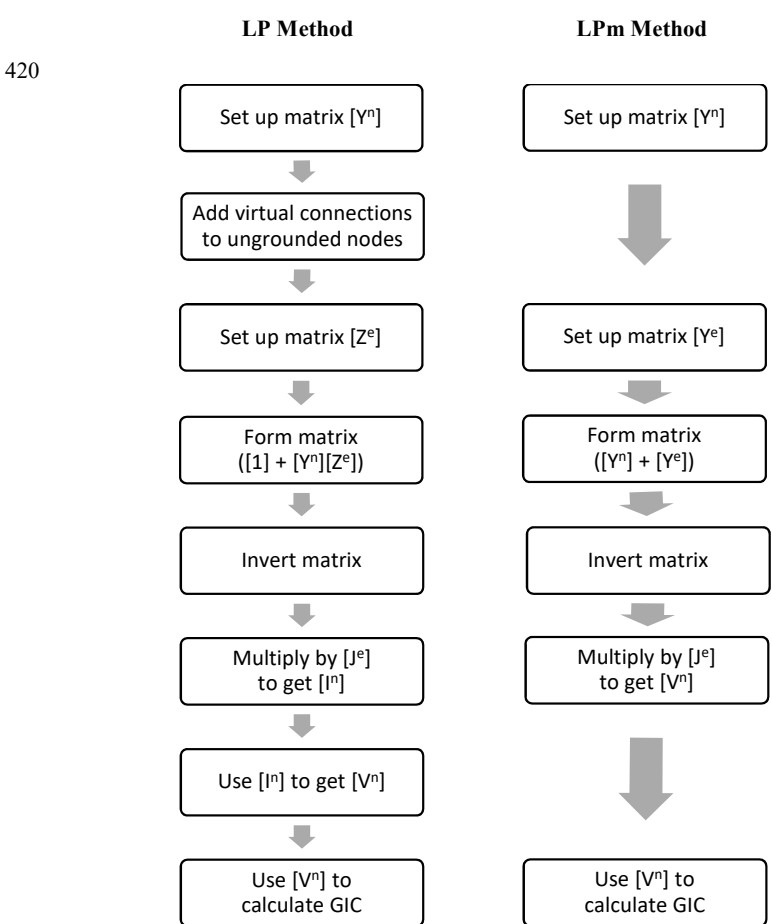

Figure 2: Comparison of the steps involved in the LP and LPm methods.




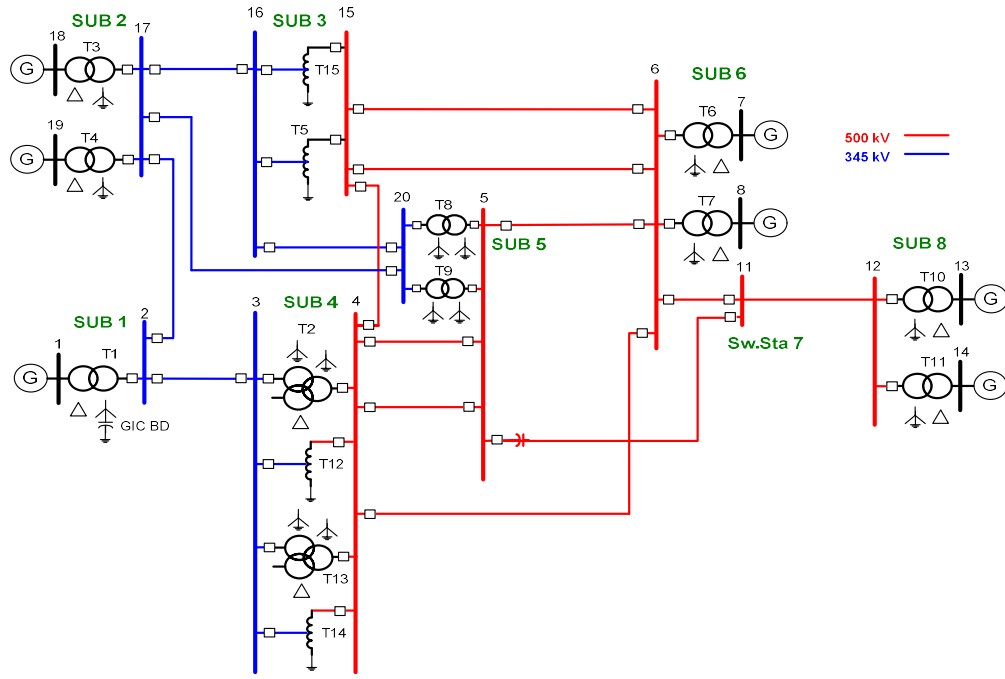

Figure 3: Single-line diagram of the benchmark test case of Horton et al. (2012).




Table 1. Assignment of node numbers.

| Substation | Location | Comment | Node |
|---|---|---|---|
| 1 | Bus 1 | delta connection | - |
| 1 | Bus 2 | | 1 |
| 2 | Bus 17 | | 2 |
| 2 | Bus 18 | delta connection | - |
| 2 | Bus 19 | delta connection | - |
| 3 | Bus 15 | | 3 |
| 3 | Bus 16 | | 4 |
| 4 | Bus 3 | | 5 |
| 4 | Bus 4 | | 6 |
| 5 | Bus 5 | | 7 |
| 5 | Bus 20 | | 8 |
| 6 | Bus 6 | | 9 |
| 6 | Bus 7 | delta connection | - |
| 6 | Bus 8 | delta connection | - |
| 7 | Bus 11 | No transformers | 10 |
| 8 | Bus 12 | | 11 |
| 8 | Bus 13 | delta connection | - |
| 8 | Bus 14 | delta connection | - |
| 1 | Neutral | Blocking device | 12 |
| 2 | Neutral | | 13 |
| 3 | Neutral | | 14 |
| 4 | Neutral | | 15 |
| 5 | Neutral | | 16 |
| 6 | Neutral | | 17 |
| 8 | Neutral | | 18 |






Table 2. Network Admittance Matrix $[Y^n]$ for the benchmark model shown in Figure 3.

| | 1 | 2 | 3 | 4 | 5 | 6 | 7 | 8 | 9 | 10 | 11 | 12 | 13 | 14 | 15 | 16 | 17 | 18 |
|---|---|---|---|---|---|---|---|---|---|---|---|---|---|---|---|---|---|---|
| 1 | 10.57 | -0.28 | 0 | 0 | -0.28 | 0 | 0 | 0 | 0 | 0 | 0 | -10 | 0 | 0 | 0 | 0 | 0 | 0 |
| 2 | -0.28 | 20.64 | 0 | -0.21 | 0 | 0 | 0 | -0.14 | 0 | 0 | 0 | 0 | -20 | 0 | 0 | 0 | 0 | 0 |
| 3 | 0 | 0 | 51.19 | -50 | 0 | -0.5 | 0 | 0 | -0.68 | 0 | 0 | 0 | 0 | 0 | 0 | 0 | 0 | 0 |
| 4 | 0 | -0.21 | -50 | 83.79 | 0 | 0 | 0 | -0.25 | 0 | 0 | 0 | 0 | 0 | -33.33 | 0 | 0 | 0 | 0 |
| 5 | -0.28 | 0 | 0 | 0 | 103.62 | -50 | 0 | 0 | 0 | 0 | 0 | 0 | 0 | 0 | -53.33 | 0 | 0 | 0 |
| 6 | 0 | 0 | -0.5 | 0 | -50 | 61.57 | -0.85 | 0 | -0.21 | 0 | 0 | 0 | 0 | 0 | -10 | 0 | 0 | 0 |
| 7 | 0 | 0 | 0 | 0 | 0 | -0.85 | 51.19 | 0 | -0.34 | 0 | 0 | 0 | 0 | 0 | 0 | -50 | 0 | 0 |
| 8 | 0 | -0.14 | 0 | -0.25 | 0 | 0 | 0 | 33.72 | 0 | 0 | 0 | 0 | 0 | 0 | 0 | -33.33 | 0 | 0 |
| 9 | 0 | 0 | -0.68 | 0 | 0 | -0.21 | -0.34 | 0 | 15.26 | -0.69 | 0 | 0 | 0 | 0 | 0 | 0 | -13.33 | 0 |
| 10 | 0 | 0 | 0 | 0 | 0 | 0 | 0 | 0 | -0.69 | 1.12 | -0.43 | 0 | 0 | 0 | 0 | 0 | 0 | 0 |
| 11 | 0 | 0 | 0 | 0 | 0 | 0 | 0 | 0 | 0 | -0.43 | 20.43 | 0 | 0 | 0 | 0 | 0 | 0 | -20 |
| 12 | -10 | 0 | 0 | 0 | 0 | 0 | 0 | 0 | 0 | 0 | 0 | 10 | 0 | 0 | 0 | 0 | 0 | 0 |
| 13 | 0 | -20 | 0 | 0 | 0 | 0 | 0 | 0 | 0 | 0 | 0 | 0 | 20 | 0 | 0 | 0 | 0 | 0 |
| 14 | 0 | 0 | 0 | -33.33 | 0 | 0 | 0 | 0 | 0 | 0 | 0 | 0 | 0 | 33.33 | 0 | 0 | 0 | 0 |
| 15 | 0 | 0 | 0 | 0 | -53.33 | -10 | 0 | 0 | 0 | 0 | 0 | 0 | 0 | 0 | 63.33 | 0 | 0 | 0 |
| 16 | 0 | 0 | 0 | 0 | 0 | 0 | -50 | -33.33 | 0 | 0 | 0 | 0 | 0 | 0 | 0 | 83.33 | 0 | 0 |
| 17 | 0 | 0 | 0 | 0 | 0 | 0 | 0 | 0 | -13.33 | 0 | 0 | 0 | 0 | 0 | 0 | 0 | 13.33 | 0 |
| 18 | 0 | 0 | 0 | 0 | 0 | 0 | 0 | 0 | 0 | 0 | -20 | 0 | 0 | 0 | 0 | 0 | 0 | 20 |

Table 3. Earthing Admittance matrix $[Y^e]$ for the benchmark model shown in Figure 3.

| | 1 | 2 | 3 | 4 | 5 | 6 | 7 | 8 | 9 | 10 | 11 | 12 | 13 | 14 | 15 | 16 | 17 | 18 |
|---|---|---|---|---|---|---|---|---|---|---|---|---|---|---|---|---|---|---|
| 1 | 0 | 0 | 0 | 0 | 0 | 0 | 0 | 0 | 0 | 0 | 0 | 0 | 0 | 0 | 0 | 0 | 0 | 0 |
| 2 | 0 | 0 | 0 | 0 | 0 | 0 | 0 | 0 | 0 | 0 | 0 | 0 | 0 | 0 | 0 | 0 | 0 | 0 |
| 3 | 0 | 0 | 0 | 0 | 0 | 0 | 0 | 0 | 0 | 0 | 0 | 0 | 0 | 0 | 0 | 0 | 0 | 0 |
| 4 | 0 | 0 | 0 | 0 | 0 | 0 | 0 | 0 | 0 | 0 | 0 | 0 | 0 | 0 | 0 | 0 | 0 | 0 |
| 5 | 0 | 0 | 0 | 0 | 0 | 0 | 0 | 0 | 0 | 0 | 0 | 0 | 0 | 0 | 0 | 0 | 0 | 0 |
| 6 | 0 | 0 | 0 | 0 | 0 | 0 | 0 | 0 | 0 | 0 | 0 | 0 | 0 | 0 | 0 | 0 | 0 | 0 |
| 7 | 0 | 0 | 0 | 0 | 0 | 0 | 0 | 0 | 0 | 0 | 0 | 0 | 0 | 0 | 0 | 0 | 0 | 0 |
| 8 | 0 | 0 | 0 | 0 | 0 | 0 | 0 | 0 | 0 | 0 | 0 | 0 | 0 | 0 | 0 | 0 | 0 | 0 |
| 9 | 0 | 0 | 0 | 0 | 0 | 0 | 0 | 0 | 0 | 0 | 0 | 0 | 0 | 0 | 0 | 0 | 0 | 0 |
| 10 | 0 | 0 | 0 | 0 | 0 | 0 | 0 | 0 | 0 | 0 | 0 | 0 | 0 | 0 | 0 | 0 | 0 | 0 |
| 11 | 0 | 0 | 0 | 0 | 0 | 0 | 0 | 0 | 0 | 0 | 0 | 0 | 0 | 0 | 0 | 0 | 0 | 0 |
| 12 | 0 | 0 | 0 | 0 | 0 | 0 | 0 | 0 | 0 | 0 | 0 | 0 | 0 | 0 | 0 | 0 | 0 | 0 |
| 13 | 0 | 0 | 0 | 0 | 0 | 0 | 0 | 0 | 0 | 0 | 0 | 0 | 1.67 | 0 | 0 | 0 | 0 | 0 |
| 14 | 0 | 0 | 0 | 0 | 0 | 0 | 0 | 0 | 0 | 0 | 0 | 0 | 0 | 1.67 | 0 | 0 | 0 | 0 |
| 15 | 0 | 0 | 0 | 0 | 0 | 0 | 0 | 0 | 0 | 0 | 0 | 0 | 0 | 0 | 0.33 | 0 | 0 | 0 |
| 16 | 0 | 0 | 0 | 0 | 0 | 0 | 0 | 0 | 0 | 0 | 0 | 0 | 0 | 0 | 0 | 3.33 | 0 | 0 |
| 17 | 0 | 0 | 0 | 0 | 0 | 0 | 0 | 0 | 0 | 0 | 0 | 0 | 0 | 0 | 0 | 0 | 3.33 | 0 |
| 18 | 0 | 0 | 0 | 0 | 0 | 0 | 0 | 0 | 0 | 0 | 0 | 0 | 0 | 0 | 0 | 0 | 0 | 3.33 |



Table 4. Inverted matrix $([Y^n] + [Y^e])^{-1}$ for the benchmark model shown in Figure 3.

| | 1 | 2 | 3 | 4 | 5 | 6 | 7 | 8 | 9 | 10 | 11 | 12 | 13 | 14 | 15 | 16 | 17 | 18 |
|---|---|---|---|---|---|---|---|---|---|---|---|---|---|---|---|---|---|---|
| 1 | 2.062 | 0.281 | 0.082 | 0.080 | 0.324 | 0.317 | 0.073 | 0.070 | 0.036 | 0.023 | 0.003 | 2.062 | 0.259 | 0.076 | 0.322 | 0.069 | 0.029 | 0.003 |
| 2 | 0.281 | 0.502 | 0.047 | 0.048 | 0.060 | 0.060 | 0.029 | 0.030 | 0.013 | 0.009 | 0.001 | 0.281 | 0.463 | 0.045 | 0.060 | 0.028 | 0.011 | 0.001 |
| 3 | 0.082 | 0.047 | 0.357 | 0.343 | 0.116 | 0.117 | 0.045 | 0.045 | 0.069 | 0.045 | 0.006 | 0.082 | 0.044 | 0.327 | 0.116 | 0.043 | 0.055 | 0.005 |
| 4 | 0.080 | 0.048 | 0.343 | 0.349 | 0.113 | 0.113 | 0.044 | 0.045 | 0.066 | 0.043 | 0.006 | 0.080 | 0.044 | 0.333 | 0.112 | 0.043 | 0.053 | 0.005 |
| 5 | 0.324 | 0.060 | 0.116 | 0.113 | 0.587 | 0.574 | 0.117 | 0.109 | 0.058 | 0.038 | 0.005 | 0.324 | 0.056 | 0.107 | 0.582 | 0.109 | 0.047 | 0.004 |
| 6 | 0.317 | 0.060 | 0.117 | 0.113 | 0.574 | 0.578 | 0.117 | 0.110 | 0.059 | 0.038 | 0.005 | 0.317 | 0.055 | 0.108 | 0.572 | 0.110 | 0.047 | 0.004 |
| 7 | 0.073 | 0.029 | 0.045 | 0.044 | 0.117 | 0.117 | 0.242 | 0.223 | 0.033 | 0.021 | 0.003 | 0.073 | 0.026 | 0.042 | 0.116 | 0.225 | 0.026 | 0.002 |
| 8 | 0.070 | 0.030 | 0.045 | 0.045 | 0.109 | 0.110 | 0.223 | 0.254 | 0.031 | 0.020 | 0.003 | 0.070 | 0.028 | 0.043 | 0.109 | 0.226 | 0.025 | 0.002 |
| 9 | 0.036 | 0.013 | 0.069 | 0.066 | 0.058 | 0.059 | 0.033 | 0.031 | 0.258 | 0.168 | 0.022 | 0.036 | 0.012 | 0.063 | 0.058 | 0.031 | 0.207 | 0.019 |
| 10 | 0.023 | 0.009 | 0.045 | 0.043 | 0.038 | 0.038 | 0.021 | 0.020 | 0.168 | 1.047 | 0.137 | 0.023 | 0.008 | 0.041 | 0.038 | 0.020 | 0.134 | 0.117 |
| 11 | 0.003 | 0.001 | 0.006 | 0.006 | 0.005 | 0.005 | 0.003 | 0.003 | 0.022 | 0.137 | 0.322 | 0.003 | 0.001 | 0.005 | 0.005 | 0.003 | 0.018 | 0.276 |
| 12 | 2.062 | 0.281 | 0.082 | 0.080 | 0.324 | 0.317 | 0.073 | 0.070 | 0.036 | 0.023 | 0.003 | 2.162 | 0.259 | 0.076 | 0.322 | 0.069 | 0.029 | 0.003 |
| 13 | 0.259 | 0.463 | 0.044 | 0.044 | 0.056 | 0.055 | 0.026 | 0.028 | 0.012 | 0.008 | 0.001 | 0.259 | 0.474 | 0.042 | 0.055 | 0.026 | 0.010 | 0.001 |
| 14 | 0.076 | 0.045 | 0.327 | 0.333 | 0.107 | 0.108 | 0.042 | 0.043 | 0.063 | 0.041 | 0.005 | 0.076 | 0.042 | 0.346 | 0.107 | 0.041 | 0.050 | 0.005 |
| 15 | 0.322 | 0.060 | 0.116 | 0.112 | 0.582 | 0.572 | 0.116 | 0.109 | 0.058 | 0.038 | 0.005 | 0.322 | 0.055 | 0.107 | 0.593 | 0.109 | 0.047 | 0.004 |
| 16 | 0.069 | 0.028 | 0.043 | 0.043 | 0.109 | 0.110 | 0.225 | 0.226 | 0.031 | 0.020 | 0.003 | 0.069 | 0.026 | 0.041 | 0.109 | 0.229 | 0.025 | 0.002 |
| 17 | 0.029 | 0.011 | 0.055 | 0.053 | 0.047 | 0.047 | 0.026 | 0.025 | 0.207 | 0.134 | 0.018 | 0.029 | 0.010 | 0.050 | 0.047 | 0.025 | 0.225 | 0.015 |
| 18 | 0.003 | 0.001 | 0.005 | 0.005 | 0.004 | 0.004 | 0.002 | 0.002 | 0.019 | 0.117 | 0.276 | 0.003 | 0.001 | 0.005 | 0.004 | 0.002 | 0.015 | 0.280 |


Table 5. Voltages in the transmission lines and equivalent current sources for northward and eastward electric fields of 1 V/km.

| Line | From Bus | To Bus | 1 V/km Northward Electric Field | | 1 V/km Eastward Electric Field | |
|---|---|---|---|---|---|---|
| | | | Vsource (V) | Jsource (A) | Vsource (V) | Jsource (A) |
| 1 | 2 | 3 | -7.28 | -2.07 | 120.60 | 34.34 |
| 2 | 2 | 17 | 77.31 | 21.93 | 93.16 | 26.43 |
| 3 | 15 | 4 | -45.16 | -22.74 | -129.27 | -65.09 |
| 4 | 17 | 16 | -39.42 | -8.45 | 155.56 | 33.35 |
| 5 | 4 | 5 | -93.47 | -39.86 | 131.69 | 56.16 |
| 6 | 4 | 5 | -93.47 | -39.86 | 131.69 | 56.16 |
| 7 | 5 | 6 | 74.56 | 25.06 | 190.99 | 64.20 |
| 8 | 5 | 11 | 171.60 | 48.90 | 169.82 | 48.40 |
| 9 | 6 | 11 | 97.05 | 67.21 | -20.14 | -13.95 |
| 10 | 4 | 6 | -18.92 | 4.05 | 321.26 | 68.85 |
| 11 | 15 | 6 | -64.08 | -21.92 | 191.11 | 65.36 |
| 12 | 15 | 6 | -64.08 | -21.92 | 191.11 | 65.36 |
| 13 | 11 | 12 | -6.29 | -2.71 | 160.17 | 68.92 |
| 14 | 16 | 20 | -138.64 | -34.24 | 1.49 | 0.37 |
| 15 | 17 | 20 | -178.06 | -25.66 | 158.17 | 22.79 |






Table 6. Nodal current sources for northward and eastward electric fields of 1 V/km.

| Node | Substation | Location | Nodal current sources (J) | |
|---|---|---|---|---|
| | | | for $E_{North}$ | for $E_{East}$ |
| 1 | 1 | Bus 2 | -19.86 | -60.77 |
| 2 | 2 | Bus 17 | 56.04 | -29.71 |
| 3 | 3 | Bus 15 | 66.57 | -65.62 |
| 4 | 3 | Bus 16 | 25.79 | 32.98 |
| 5 | 4 | Bus 3 | -2.07 | 34.34 |
| 6 | 4 | Bus 4 | 61.03 | -246.26 |
| 7 | 5 | Bus 5 | -104.78 | 48.12 |
| 8 | 5 | Bus 20 | -59.90 | 23.16 |
| 9 | 6 | Bus 6 | -90.03 | 277.71 |
| 10 | 7 | Bus 11 | 69.91 | -82.87 |
| 11 | 8 | Bus 12 | -2.71 | 68.92 |
| 12 | 1 | Neutral | 0.00 | 0.00 |
| 13 | 2 | Neutral | 0.00 | 0.00 |
| 14 | 3 | Neutral | 0.00 | 0.00 |
| 15 | 4 | Neutral | 0.00 | 0.00 |
| 16 | 5 | Neutral | 0.00 | 0.00 |
| 17 | 6 | Neutral | 0.00 | 0.00 |
| 18 | 8 | Neutral | 0.00 | 0.00 |

Table 7. Nodal voltages produced by northward and eastward electric fields of 1 V/km.

| Node | Substation | Location | Node voltages (V) | |
|---|---|---|---|---|
| | | | for $E_{North}$ | for $E_{East}$ |
| 1 | 1 | Bus 2 | -12.39 | -190.04 |
| 2 | 2 | Bus 17 | 25.05 | -41.01 |
| 3 | 3 | Bus 15 | 30.09 | -24.39 |
| 4 | 3 | Bus 16 | 29.37 | -22.99 |
| 5 | 4 | Bus 3 | 20.04 | -125.10 |
| 6 | 4 | Bus 4 | 20.33 | -125.97 |
| 7 | 5 | Bus 5 | -29.01 | -7.26 |
| 8 | 5 | Bus 20 | -29.04 | -6.13 |
| 9 | 6 | Bus 6 | -7.16 | 44.32 |
| 10 | 7 | Bus 11 | 60.57 | -40.47 |
| 11 | 8 | Bus 12 | 7.11 | 15.67 |
| 12 | 1 | Neutral | -12.39 | -190.04 |
| 13 | 2 | Neutral | 23.13 | -37.86 |
| 14 | 3 | Neutral | 27.97 | -21.90 |





| 15 | 4 | Neutral | 19.98 | -124.58 |
|----|---|---------|-------|---------|
| 16 | 5 | Neutral | -27.91 | -6.55 |
| 17 | 6 | Neutral | -5.73 | 35.45 |
| 18 | 8 | Neutral | 6.09 | 13.43 |


Table 8. Properties of the matrices to be inverted using the LP and LPm methods for different power networks, namely Horton et al. (2012) benchmark and REE.

| Model | Method | Matrix Size | Inversion Process | Storage (kB) | Inversion Time (µs) |
|-------|--------|-------------|-------------------|--------------|---------------------|
| Horton | LP | 18 x 18 | Regular | 2.6 | 12 |
| | LPm | 18 x 18 | Regular | 2.6 | 11 |
| REE | LP | 1388 x 1388 | Regular | $15.4 \cdot 10^3$ | $6 \cdot 10^4$ |
| | LPm | 1388 x 1388 | Regular | $15.4 \cdot 10^3$ | $4 \cdot 10^4$ |
| Horton | LP | 18 x 18 | Sparse | 1.3 | 90 |
| | LPm | 18 x 18 | Sparse | 1.2 | 14 |
| REE | LP | 1388 x 1388 | Sparse | 110 | $7 \cdot 10^3$ |
| | LPm | 1388 x 1388 | Sparse | 90 | $9 \cdot 10^2$ |
