# Peer review of "The Lehtinen-Pirjola Method Modified for Efficient Modelling of Geomagnetically Induced Currents in Multiple Voltage Levels of a Power Network"

_Annales Geophysicae, 2021_

## Author Response (AR1)

**Response to the Reviews**

**Response to Comments by Referee 1 (in bold)**

This paper provides a new method for computing GIC by simplifying the manner in which grounding resistances are treated in the traditional Lethinen and Pirjola (1985) paper. The authors revisit the derivations of the LP85 paper and from there explain their suggested change for removing virtual nodes from the equations, which are usually given a very large (e.g. 10,000 Ohm) value to represent an infinite resistance to ground. This is a useful update and easy to implement. The main advantage would be if you were to start afresh though existing codes could also be modified. The authors test the new implementation agains the Horton et al (2012) paper and reproduce the results given in the original Horton et al paper. This is a very pleasing demonstration of the method and a very useful check for others attempting to develop their own code.

**We appreciate the comment that this method is useful and easy to implement. The referee notes that the main advantage would be for new code but acknowledges that existing codes could also be modified. We would like to stress the value of the new method for existing codes. Fig 2 compares the steps involved in the LP method and the LPm method and shows that the changes to existing code are minimal. Text has been added in section 5 to explain the changes that are needed. Therefore, we hope that existing LP code will be modified to use the LPm method.**

Comments:

1) The use of LU decomposition is quite specific. It was not clear if the authors directly compute the inverse of [Y^e + Y^n] using Matlab's pinv() function. This would not be the correct way to solve it as Matlab usually finds the best way to resolve the inversion if the A\b form (i.e. the backslash function) was used. Can I ask how the values in Table 8 were arrived at? If you are computing the inverse directly, that is not really a fair comparison of potential compute time or the savings from LP versus LPm.

**We use the following Matlab commands for LPm:**

**decompM = decomposition(M, 'Chol');**
**Vdg = decompM\Je; % Nodal voltages.**

**The first gives the Cholesky decomposition of matrix M, which is feasible because M is symmetric positive definite\*; the second line performs the forward and back substitutions on the decomposition object decompM.**

**While the commands for LP are:**

**decompM = decomposition(Mred);**
**Ig = decompM\Je; % Nodal GIC to ground.**

In this case, the first line gives the most efficient decomposition of matrix M*; as above, the second line performs the forward and back substitutions on the decomposition object decompM.

* According to Matlab documentation, decomposition command "*creates reusable matrix decompositions (LU, LDL, Cholesky, QR, ...) that enable to solve linear systems (Ax = b or xA = b) more efficiently. For example, after computing dA = decomposition(A) the call dA\b returns the same vector as A\b but is typically much faster. decomposition objects are well-suited to solving problems that require repeated solutions, since the decomposition of the coefficient matrix does not need to be performed multiple times.*" Note that GIC computation problems often require repeated solutions as the geoelectric field changes with time (while M remains the same), which justifies performing the decomposition.

Irrespective of all the above, we would like to note a mistake in 3 of the time values reported for the Horton et al. (2012) benchmark circuit in Table 8: those times account only for the second command in both cases (i.e., the forward and back substitutions). In contrast, in lines 295 - 296 of the manuscript we state that the reported times include both, the decomposition + the forward and back substitutions (i.e., the two commands). In consequence, the correct times should indeed be greater, though our arguments remain the same. Namely, the last column should be changed to:

| Inversion Time (µs) |
| --- |
| 12 |
|  10 |
| $6·10^4$ |
| $4·10^4$ |
|  320 |
|  170 |
| $7·10^3$ |
| $9·10^2$ |

**Theses changes have been made in the revised manuscript.**

2) I have found in the standard LP method that when grounding resistance is high (10,000 Ohm) the current in virtual nodes is usually near zero to within 5 decimal places. So altough this modified LPm method is mathemtically better, it is not usually an issue with regards to computing the 'wrong' value, as any GIC below 0.001 A can be considered to be zero in reality.

**We agree that a GIC value below 0.001 A can be considered to be zero in reality. But our concern is not with regard to computing the wrong value for the GIC flow to**

**ground from the virtual nodes. Our concern is that, in a system with multiple voltage levels, it is necessary to calculate the nodal voltages first and then use these to calculate the GIC in the transformer windings. With the use of virtual nodes in the LP method the nodal voltage ends up being calculated by multiplying the very high resistance to ground by the very small current flow to ground. Any round off errors in the small current value are then going to introduce errors in the value obtained for the nodal voltage.**

3) Line 225: Yn should be $Y^n$ (twice)

**Corrected**

4) In the contributions, I assume AM is SM?

**Yes. Corrected**

**Response to Comments by Referee 2 (in bold)**

**We appreciate the positive comments about the paper.**

**Response to Specific Comments**

Line 34: Suggest including "and space weather" after "geophysics".

**Suggested text added.**

Page 3, line 1: Perhaps "relate" should be "relates".

**Change made.**

Page 4 and throughout: In some instances, the matrix notation is inline (e.g., $[I_e]$ in line 97) and other times it is raised/split above/between lines (e.g., $[I_e]$ in line 106). Perhaps a consistent notation should be used throughout.

**The raised notation was a mistake. This has been corrected.**

Line 154: Appears to use different font for equation 22 (compare with equation 16 for example).

**The font for the equations has been standardised.**

Line 282: Remove space before comma and end of line.

**Done.**

Lines 337-338: "The research that led to these results was in part carried out using funds from "la Caixa" Foundation. Natural Resources Canada contribution number 20210276", seems to require a comma separating the sentences?

**Text has been re-arranged.**

Table 5: Formatting of "To/From Bus" column headings perhaps needs adjusting.

**Done.**

Tables 8: Formatting of last 3 column headings perhaps needs adjusting.

**Done.**